# FINDING WINNING TICKETS WITH LIMITED (OR NO) SUPERVISION

## ABSTRACT

The lottery ticket hypothesis argues that neural networks contain sparse subnetworks, which, if appropriately initialized (the *winning tickets*), are capable of matching the accuracy of the full network when trained in isolation. Empirically made in different contexts, such an observation opens interesting questions about the dynamics of neural network optimization and the importance of their initializations. However, the properties of winning tickets are not well understood, especially the importance of supervision in the generating process. In this paper, we aim to answer the following open questions: can we find winning tickets with few data samples or few labels? can we even obtain "good" tickets without supervision? We find that winning tickets found in these scenarios are, perhaps surprisingly, competitive with winning tickets generated on the full ImageNet dataset when evaluated on ImageNet classification task.

## 1 INTRODUCTION

Recently, Frankle & Carbin (2019) have observed that sparse subnetworks of over-parameterized neural networks could achieve good predictions when trained in isolation, as long as they are appropriately initialized; these "lucky" starting points have been termed *winning tickets*. Building these tickets is typically achieved by pruning the weights with lowest magnitude of an over-parametrized network that has been trained to convergence, before resetting the remaining weights to their initial values, or at some point early in training, and repeating the procedure. The properties of these effective subnetworks and their initialization are however not well understood yet. For example, the original lottery ticket hypothesis argued that these winning tickets emerge from the network initialization. Yet, original experiments were performed on classification tasks with shallow architectures and small datasets, and further experiments on deeper models and more challenging problems suggest that they may rather appear later, though still early, during training (Frankle et al., 2019). Recent works (Morcos et al., 2019; Yu et al., 2019; Zhou et al., 2019) have studied other properties of the winning tickets generation process: Morcos et al. (2019) have shown that winning tickets can be transferred across datasets with similar natural image statistics, and Yu et al. (2019) have exposed the existence of lottery tickets to other domains, such as text and reinforcement learning.

In this paper, we expand on this line of work and empirically answer other important open questions on winning tickets and the data-labels distribution on which they are generated. In particular, we want to know if "good" tickets can be obtained when few data samples, or few labels, or even no labels, are available. Answering these questions is important since it may speed-up the winning ticket generation process, which is computationnally expensive, and also open new perspectives about generating sparse subnetworks that may be trained efficiently on new tasks.

To this effect, we design several experiments that isolate the impact of the data and label distributions on the quality of the resulting winning tickets. First, we analyze the extent to which winning tickets are label-dependent by generating "label-agnostic" winning tickets with self-supervised tasks (Gidaris et al., 2018; Doersch & Zisserman, 2017). Then, we evaluate the impact of reducing the numbers of samples per class or the number of classes used for winning ticket generation. Finally, we evaluate the performance of semi-supervised approaches to winning ticket generation. Most of our experiments are conducted on ImageNet; we remark indeed that deep networks trained on smaller

datasets such as CIFAR-10 are already sparse at convergence, making pruning less challenging, and conclusions drawn about lottery tickets potentially misleading if this effect is not accounted for.

Overall, our experiments show that winning tickets are surprisingly robust to many of these data/label distribution changes. Indeed, using only $10\%$ of the dataset or removing entirely the labels still leads to effective winning tickets. As a result, we manage to extract winning tickets $5\times$ faster by training on a subset of data, with no modifications to the core algorithm. This is of particular interest because finding winning tickets is highly data, labels and resource-demanding since it requires training a network to convergence at least once (and generally dozens of times in practice).

Our paper makes the following contributions: **(i)** We combine self-supervised learning with winning tickets generation, showing that good winning tickets can be found without labels. **(ii)** We show that finding winning tickets can be accelerated by a factor $5\times$ on ImageNet by using only a subset of the data. **(iii)** We also show that using large datasets is important to study lottery tickets, since deep networks trained on CIFAR-10 are natually sparse, making conclusions potentially misleading.

## 2 RELATED WORK

**Pruning.**    Pruning is an approach to model compression (Han et al., 2015) and regularization (Le-Cun et al., 1990) in which weights or nodes/filters are removed, typically by clamping them to zero. It is an active research which has primarily focused on pruning an already trained network (Han et al., 2015; Li et al., 2017) or pruning while training (Prakash et al., 2019). In particular, the iterative pruning during training of Han et al. (2015) has been extended to continuous pruning (Guo et al., 2016), layer-wise pruning (Dong et al., 2017) and with weight sharing (Ullrich et al., 2017). Pruning during training has been considered with $\ell_0$ regularization (Louizos et al., 2018), binary convolution (Rastegari et al., 2016) or using the hashing trick for weight sharing (Chen et al., 2015). There are several ways to prune a network; among them are unstructured pruning (LeCun et al., 1990; Han et al., 2015) which corresponds to discarding *individual* connections or structured pruning that operates by removing entire channels or layers (see Liu et al. (2019) for a review of the different methods). The lottery ticket hypothesis focuses on unstructured pruning early in training, which has the advantage of significantly reducing the training cost of the resulting model.

**The lottery tickets hypothesis.**    The lottery tickets hypothesis (Frankle & Carbin, 2019) explores the possibility of pruning early in training by revealing that some sparse subnetworks inside neural networks can reach accuracy matching that of the full network when trained in isolation. Setting the weights of the sparse architecture appropriately is critical to reach good performance. Frankle & Carbin (2019) provide a proof of concept of the lottery ticket hypothesis on small vision benchmarks while Frankle et al. (2019) conduct further experiments with deeper networks, which result in the introduction of rewinding and a consequent revision to the original hypothesis. Liu et al. (2019) question the importance of weights resetting and observe for moderate pruning rates and without rewinding that the pruned architecture alone is responsible for successful training. Zhou et al. (2019) conduct ablation studies on the lottery tickets hypothesis and show, among others, the importance of the signs of the reset weights. Similar to our paper, these works aim at better understanding winning tickets properties; however none of them investigate their relationship with the data and labels on which they are trained, though Yu et al. (2019) investigate lottery tickets in reinforcement learning problems. Closer to our work, Morcos et al. (2019) show that winning tickets initializations transfer across different image classification datasets, thus suggesting that winning tickets do not entirely overfit to the particular data distribution on which they are found. Here, we follow up on these observations by measuring the importance of labels and data size to the quality of winning tickets.

**Learning without supervision.**    As we study situations with limited access to labels, our work is also related to self-supervised learning where a network is trained on a pretext task that does not require any manual annotations. Two main broad types of self-supervised learning approaches appear in the litterature. The first one consists of methods where the pretext task is created by manipulating the input data. This includes predicting relative spatial location, colorizing grayscale images or predicting the rotation applied to an image (Doersch et al., 2015; Wang & Gupta, 2015; Zhang et al., 2016; Noroozi & Favaro, 2016; Pathak et al., 2017; Gidaris et al., 2018; Tian et al., 2019; Hénaff et al., 2019). The second one is composed of methods (Dosovitskiy et al., 2016; Bojanowski & Joulin, 2017; Wu et al., 2018; Caron et al., 2018) where images are treated as different instances

that should be discriminated from one another. Representations learnt using self-supervision are most often evaluated via transfer learning to a supervised task. The better the pre-training with self-supervised learning, the better the performance on the transfer task. In this work, we propose a new way of exploiting signal learned with self-supervision by findind winning tickets.

## 3 Background on Lottery Tickets and Implementation

In this section, we provide some background on the lottery tickets hypothesis and more particularly about the procedure to find winning tickets. We also give more details about our implementation.

### 3.1 Extracting and training subnetworks

We detail in this section the approach of Frankle et al. (2019) to extract sparse and trainable subnetworks from an over-parameterized network. These subnetworks, or winning tickets, are generated with a magnitude-based unstructured iterative pruning process (Han et al., 2015) and a weight resetting scheme (Frankle & Carbin, 2019; Frankle et al., 2019).

**Subnetwork and dataset.** A subnetwork or winning ticket $(W, m)$ is represented by the association of a mask $m$ in $\{0, 1\}^d$ and weights $W$ in $\mathbb{R}^d$. The convolutional network, or convnet, function associated with a subnetwork $(W, m)$ is denoted by $f_{m \odot W}$, where $\odot$ is element-wise product. We consider a training set $\{(x_1, y_1), \ldots, (x_N, y_N)\}$ of $N$ pairs of images and their labels.

**Magnitude-based unstructured iterative pruning.** Han et al. (2015) propose an algorithm to prune networks by estimating which weights are important. This approach consists of compressing networks by alternatively minimizing a loss $\mathcal{L}_g$ and pruning the network parameters with the smallest magnitude, hence progressively reducing the network size. We refer to the training objective $\mathcal{L}_g$ as the winning ticket generation task. More specifically, at each pruning iteration, the network is first trained to minimize the following problem:

$$\min_W \frac{1}{N} \sum_{n=1}^N \mathcal{L}_g(y_n, f_{m \odot W}(x_n)), \tag{1}$$

thus arriving at weights $W^*$. The mask $m$ is updated by setting to zero the elements already masked plus the smallest elements of $\{|W^*[j]| \,|\, m[j] \neq 0\}$. This method operates as a post-processing step to prune an already-trained network while preserving good accuracy at high compression rates.

**Weight resetting.** In addition to weight pruning, Frankle & Carbin (2019) propose to also find a good initialization for each subnetwork, such that it may be retrained from scratch. On small-scale computer vision datasets and with shallow architectures, they indeed show that sub-architectures found with iterative magnitude pruning can be retrained from the start, as long as their weights are reset to their initial values. Further experiments, however, have shown that this observation does not exactly hold for more challenging benchmarks such as ImageNet (Frankle et al., 2019; Liu et al., 2019). Specifically, Frankle et al. (2019) found that resetting weights to their value "early in training" can still lead to good trainable subnetworks. Formally, at each pruning iteration, the subnetwork is reset to weights $W_k$ obtained after $k$ weight updates from the first pruning iteration.

**Winning ticket generation task.** Most studies about the lottery tickets hypothesis evaluate winning tickets in the same setting (i.e. task and dataset) used to find them. In other words, the resulting winning tickets performance is measured by re-training them on the same objective $\mathcal{L}_g$ that was used in the generation process (Eq. 1). In this paper, however, we vary the generation loss $\mathcal{L}_g$ while the evaluation remains constant. For example, to generate "label-agnostic" winning tickets (i.e. that do not depend on labels), $\mathcal{L}_g$ might correspond to a self-supervised loss. We evaluate winning tickets by reporting their performance after training on label classification on the full dataset.

### 3.2 Implementation

**Lottery tickets.** We follow the lottery tickets setup of Morcos et al. (2019). At each pruning iteration, we globally pruned 20% of the remaining weights. The last fully-connected and batch-

norm layers parameters are left unpruned. We apply up to 30 pruning iterations to reach extreme pruning rates where only $0.1\%$ of the weights remain. Late resetting is set to $3 \times 1, 3\text{M}$ samples, which corresponds to 3 epochs on full ImageNet. More details about the rewinding parameter are in the Appendix B. We use the "random" baseline of Morcos et al. (2019) (random mask and weights).

**Datasets and models.** We use CIFAR-10 (Krizhevsky, 2009) or ImageNet (Deng et al., 2009) datasets. On ImageNet, we consider AlexNet (Krizhevsky et al., 2012) and ResNet-50 (He et al., 2016) architectures. On CIFAR-10 we use ResNet-18 and the modified VGG-19 (Simonyan & Zisserman, 2014) of Morcos et al. (2019) (multilayer perceptron (MLP) is replaced by a fully connected layer). Models are trained with an $\ell_2$ penalization of the weights. Details about optimization schemes are in Appendix A. We run each experiment with 6 (CIFAR-10) or 3 (ImageNet) random seeds, and report the mean and standard error of the accuracy (tables of results are in Appendix E).

## 4 EXPERIMENTAL STUDY

In this section, we look at how "label-agnostic" winning tickets transfer to label-dependent task. Then, we investigate if we can find good performing winning tickets with a reduced number of samples and/or classes. Finally, we also question if the "label-agnostic" winning tickets are complementary to the tickets found on a subset only of data through a semi-supervised generation task.

### 4.1 LABEL-AGNOSTIC WINNING TICKETS

In this section, we are interested in the properties of "label-agnostic" winning tickets, i.e. found with self-supervised tasks, w.r.t. to a target supervised task. We compare their performance with winning tickets found with supervision to measure the importance of labels in the winning tickets generation.

**Experimental setting.** We consider 2 self-supervised methods: RotNet (Gidaris et al., 2018) and the Exemplar approach of Dosovitskiy et al. (2016) following Doersch & Zisserman (2017) implementation. RotNet consists in predicting the rotation which was applied to the input image among a set of 4 possible large rotations: $\{0°, 90°, 180°, 270°\}$. Exemplar is a classification problem where each image and its transformations form a class, leading to as many classes as there are training examples. We choose these 2 self-supervised tasks because they have opposite characteristics: RotNet encourages discriminative features to data transformations and has a small number of classes, while Exemplar encourages invariance to data transformations and its output space dimension is large.

**Transferring label-agnostic winning tickets to a supervised task.** In Figure 1, we show the performance on label classification of the label-agnostic winning tickets at different pruning ratios. These tickets yield good accuracy on label classification on both CIFAR-10 and ImageNet datasets. Indeed, on CIFAR-10 (Figure 1a), when only $6.9\%$ of the weights remain, the RotNet winning ticket performs as well as the labels winning ticket. The label-agnostic tickets only fail for AlexNet where most of the weights are in the MLP used in the classifier. An explanation is that, since the MLP corresponds to the last layers of the network, it is greatly influenced by the labels. We investigate this hypothesis in the set of experiments below. Note that, however, on all the other models, the gap of performance between label-agnostic tickets and random masks increases when the network is severely pruned. In the extreme pruning regime, the label-agnostic winning tickets perform as well as the ones generated with the supervised task. Results in Appendix G suggest that the quality of the pruned mask itself is similar for labels and self-supervised subnetworks but the *weights* of self-supervised winning tickets are not as good starting points as the ones from labels. Overall, winning tickets are rather robust to the absence of labels.

**Layerwise lottery tickets.** We further investigate the difference between label-agnostic and supervised lottery tickets in AlexNet by looking at their performance as we prune up to a given depth. In Figure 2, we verify that for the convolutional layers, the label-agnostic tickets match the performance of supervised tickets, and the performance drops only with the MLP. Note that, even if this effect is particularly visible with an AlexNet, it is happening with most self-supervisedly trained network. Indeed, it has been observed that most self-supervised approaches (Jing & Tian, 2019) produce good shallow and mid-level features but poor high level features. This means that we should expect the

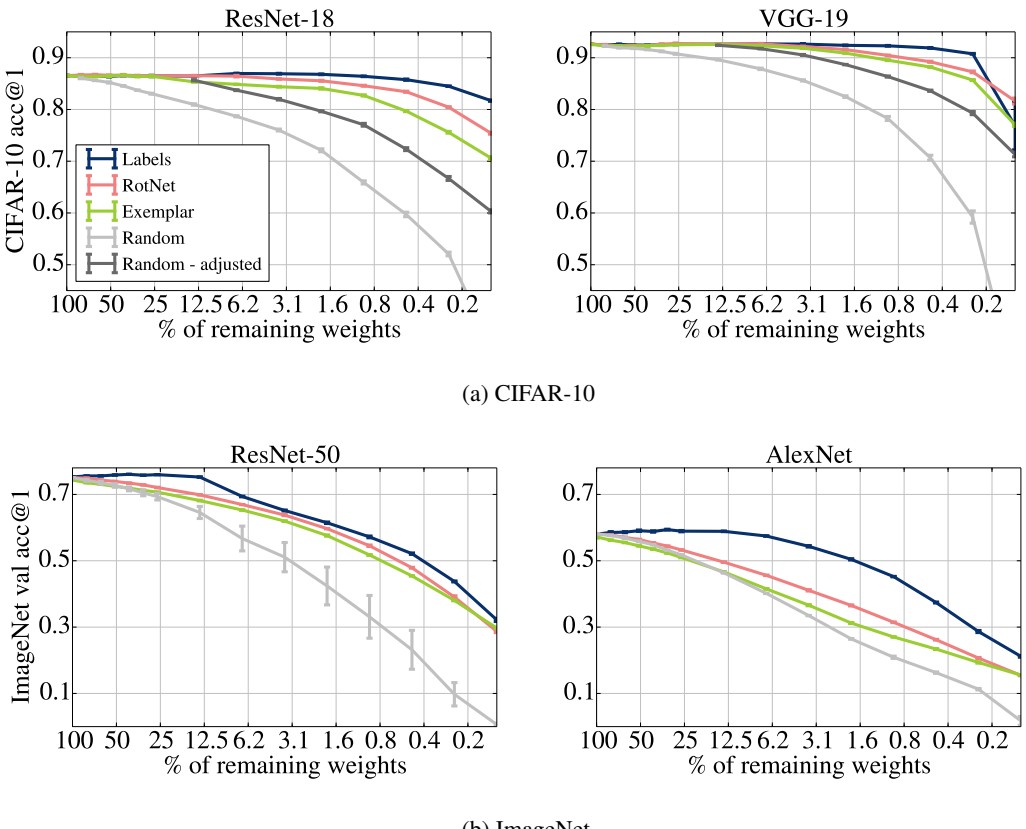

(a) CIFAR-10

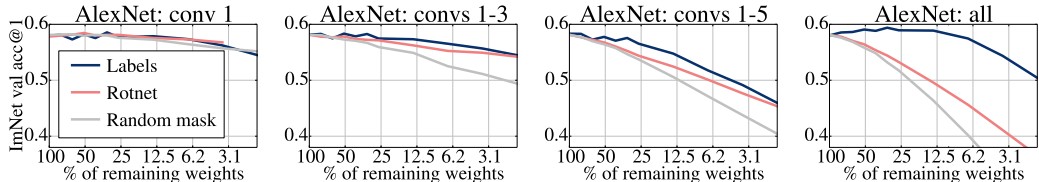

(b) ImageNet

Figure 1: We report CIFAR-10 test (b) and ImageNet val (a) top-1 accuracy for winning tickets found with label-agnostic tasks: RotNet or Exemplar. The x-axis corresponds to different pruning ratios. We compare the performance with supervised winning tickets and random subnetworks (randomly drawn weights from the initialization distribution and randomly permuted masks). On CIFAR-10, deep models are highly sparse with only $\sim 15\%$ of non-zero weights. Thus, we adjust the random baseline to start with the correct mask at the natural level of network sparsity (4.5 for details).

Figure 2: ImageNet top-1 validation accuracy of AlexNet winning tickets generated by pruning partly or entirely (all) a network with 2 generation tasks: labels classification or RotNet. We also show for reference results when the network layers are randomly pruned.

quality of label-agnostic winning tickets to depend on the depth of the pruning. In Appendix C, we confirm this intuition with a ResNet-50, and show the difference between label-agnostic and supervised lottery tickets increases with depth.

## 4.2 LOW-SHOT WINNING TICKETS

In this section, we are interesting in the quality of winning tickets generated with only a few annotated images. We explore several ways to extract a subset by varying its size or its distribution and study the impact of these choices on the subsequent winning tickets quality.

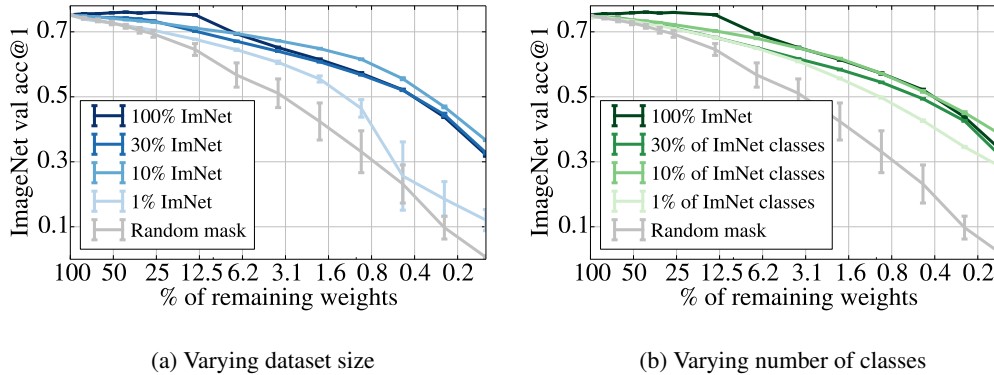

(a) Varying dataset size          (b) Varying number of classes

Figure 3: Top-1 accuracy on ImageNet validation set of winning tickets generated with different training data. The x-axis represents different levels of winning ticket sparsity.

**Experimental setting.** We consider either $1\%$, $10\%$ or $30\%$ of ImageNet, with $1\%$ corresponding to approximately 13k images. When varying the dataset size, we sample the subset per class, i.e., we preserve all the classes. When varying the number of classes, we sample a subset of classes and keep all the images for these classes, i.e., $10\%$ of ImageNet means that we only have 100 classes.

**Varying dataset size.** First, we generate winning tickets using a subset of the training data while preserving the same number of classes. Results for CIFAR-10 are in Appendix F. Note that generating winning tickets with $10\%$ of ImageNet takes $5\times$ less time than with the full dataset. In Figure 3a, we show the ImageNet validation accuracy for the winning tickets of a ResNet-50 generated with these different subsets. In less extreme pruning scenarii, there is a gap of $1 - 4\%$ between winning tickets generated with the full training set and $10\%$ of it but the winning tickets behave in the same ballpark for all settings for extreme pruning, i.e., for more than $90\%$ of weights pruned. This experiment shows that winning tickets are not data hungry, especially for extreme pruning.

**Varying the number of classes.** In Figure 3b, we analyze the impact of changing the joint data-label distribution while keeping the subsets size constant with that of Figure 3b. Specifically, we are interested in the impact of the number of classes used during the generation of the tickets. We observe that reducing the number of labels has little effect on the quality of the winning tickets. This observation reinforces the findings on label-agnostic generation, that winning tickets are robust to the absence or deterioration of the labels. Even more surprisingly, we observe good performance with only $1\%$ of the classes, i.e., 10 classes and $13k$ images, leading to a $10\times$ faster generation.

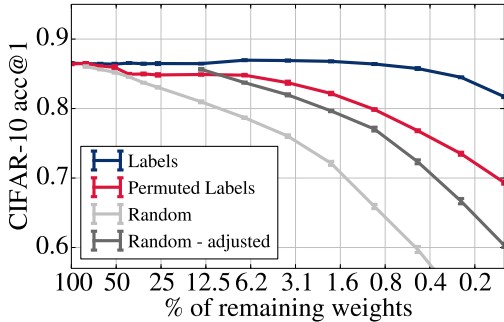

Figure 4: CIFAR-10 top-1 test accuracy of winning tickets generated by labels classification on a randomly permuted labels. Deep models are highly sparse on CIFAR-10 with only $15\%$ of non-zero weights (see 4.5 for details). For this reason we adjust the random baseline to start with the correct mask for $85\%$ sparsity.

**Decorrelating label and image distributions.** A possible explanation for these positive results about label-agnostic and low-shot winning tickets is that winning tickets generated with any task related to the right target distribution lead to good tickets. As a control experiment, we propose

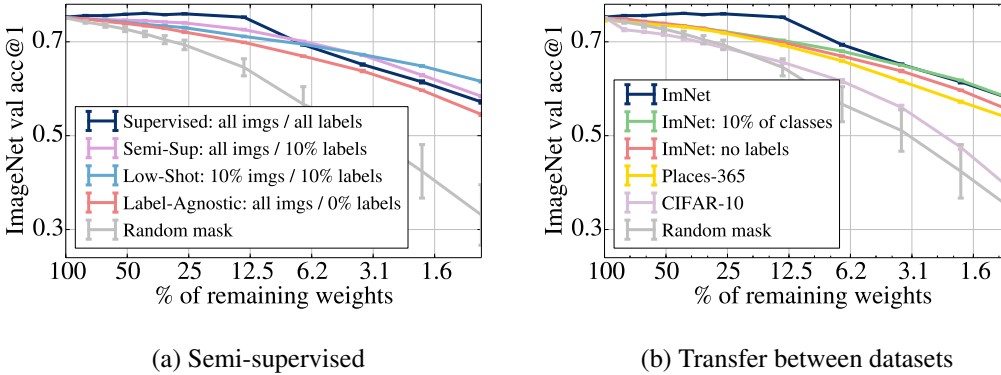

(a) Semi-supervised        (b) Transfer between datasets

Figure 5: (a) ImageNet top-1 validation accuracy of winning tickets generated with different access to both data and labels. We compare using the full dataset, all images but only $10\%$ of labels (semi-supervised), $10\%$ of labels only and all images but no labels (self-supervised). (b) ImageNet top-1 validation accuracy of winning tickets generated from other datasets.

to learn winning tickets on a classification tasks where the labels have been shuffled, i.e., where the image and label distributions have been decorrelated. In Figure 4, we show on CIFAR-10 the performance of winning tickets generated on randomly permuted labels. As expected, decorrelating the label and the image distributions destroys the quality of the subsequent winning tickets, since they are not better than the adjusted random baseline (see Sec. 4.5 for details on this baseline).

### 4.3 Semi-supervised winning tickets

In this experiment, we investigate if label-agnostic and low-shot winning tickets are complementary.

**Experimental setting.** We find winning tickets by training networks in the semi-supervised setting of Zhai et al. (2019) with $10\%$ of labeled images from ImageNet and the remaining $90\%$ unlabeled. In particular, the generation task $\mathcal{L}_g$ corresponds to the sum of a label classification loss applied to labeled examples only, and a RotNet loss applied to all data samples.

**Results.** In Figure 5-a, we compare these winning tickets with the label-agnostic, the low-shot and the supervised ones. Surprisingly, the low-shot tickets perform better than the label-agnostic ones. More importantly, we observe that winning tickets generated in the semi-supervised setting perform better in general than both the label-agnostic and low-shot ones. This shows that properties of both appraoches capture different statistics that add up to generate better winning tickets.

### 4.4 Comparaison with winning tickets transferred from other datasets

In this experiment, we compare the winning tickets transferred from other datasets to those obtained without labels or limited number of data.

**Experimental setting.** We follow the setting of Morcos et al. (2019) and use Places365 (Zhou et al., 2017) and CIFAR-10 which count respectively $1.8M$ and $50k$ training data samples and 365 and 10 categories. We consider these two datasets because they have different statistics, in particular Places365 is very large while CIFAR-10 is of comparable size with the subsets used for our low-shot experiments. For both datasets, we generate winning tickets with supervision.

**Results.** In Figure 5-b, we compare these lottery tickets with those transferred from other data distributions even if trained with more data. Interestingly, winning tickets generated with only $10\%$ of the images perform better than the tickets generated on Places365 with $14\times$ more data and $3\times$ more classes. This suggests that lottery tickets are more sensitive to the data distribution than the number of samples or the use of labels to generate them.

### 4.5 CAVEAT ABOUT PRUNING DEEP NETWORKS ON CIFAR-10

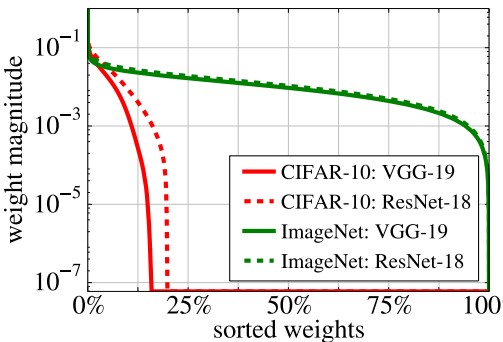

Figure 6: Magnitude of the weights of a trained network on two different datasets: CIFAR-10 (green) and ImageNet (red). We perform thresholding at machine precision value (bottom of y-axis). On CIFAR-10, a trained VGG-19 is $84.5\%$ sparse while a trained ResNet-18 is $80.3\%$ sparse.

Somewhat surprisingly, we find that prior to any pruning, a large proportion of the weights of a deep architecture trained on CIFAR-10 has converged naturally to zero during training. For example, we observe in Figure 6 that $\sim 85\%$ of the weights of a VGG-19 and $\sim 80\%$ of that of a ResNet-18 at convergence are zeroed (Appendix D shows results on more architectures). As a result, it is trivial to prune these networks without any loss in accuracy. Unstructured magnitude-based pruning acts here as *training* since are freezed to zero weights that were going to zero anyway (Zhou et al., 2019). Overall, while pruning on CIFAR-10 large networks originally tuned for ImageNet at rates above their natural level of sparsity ($\sim 80\%$) is still meaningful, analyzing pruning below this rate may not be conclusive. We find that in the random global pruning baseline (which can remove non zero weights) in Figure 1a, pruning at rates below the natural sparsity of the network degrades accuracy, while pruning of weights that are already zeroed has no effect. Inconveniently, this performance gap carries over to higher pruning rates (in which we are interested in) and can lead to misleading interpretations. For fair comparison, we adjust the random mask baseline in Figure 1a: we remove this effect by first pruning the weights that naturally converge to zero after training. Then, we randomly mask the remaining non-zeroed weights to get different final desired pruning rates. The remaining non-masked weights are randomly initialized. This baseline therefore corrects for the natural sparsity present in CIFAR-10 networks.

## 5 DISCUSSION

Overall, our study empirically shows that we can find winning tickets with good accuracy on label classification with no supervision at all. This observation suggests that winning tickets are, to some extent, surprisingly labels and task-independent. We also find that they are not very data-demanding since a dataset subset is sufficient to generate them. Finding winning tickets with a subset of ImageNet is $5\times$ faster than using the entire dataset, without modifying the original algorithm. Such a speedup is especially crucial for extreme pruning rates that are computationally demanding.

Nonetheless, we find several critical limitations to winning tickets in our study. First, none of the tickets found with limited access to labels and or data matches the accuracy of tickets found with all the labeled data when considering moderate pruning rates (more than $\sim 10\%$ of unpruned weights) on ImageNet. Indeed, we consistently observe a decrease in performance compared to the full over-parametrized network as soon as we prune the network. Therefore, for these rates, winning tickets are certainly label and data dependant. Second, we find that pruning large modern architectures on CIFAR-10 should be done with caution as these networks tend to be sparse at convergence, making unstructured pruning at rates below $80\%$ particularly simple. Third, we observe that winning tickets are particularly sensitive to the late resetting parameter (see Appendix B for a discussion about our choice of rewind parameter). The definition of "early in training" is somehow ill-defined: network weights change much more for the first epochs than for the last ones. Thus, by resetting weights early in their optimization, they still contain a vast amount of information.

Finally, we have observed winning tickets through the scope of label classification for images with natural-looking statistics only. Considering more self-supervised tasks and even different types of tasks (generative modeling for example) might complement furthermore this study.

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

## 6 APPENDICES

### A HYPERPARAMETERS AND MODEL DETAILS

We detail in this appendix the different hyperparameters used in our experiments. For all trainings on ImageNet, we use stochastic gradient descent with a momentum of 0.9. We use Adam optimizer on CIFAR-10. On ImageNet, unless specified otherwise, we perform standard data augmentation consisting in croppings of random sizes and aspect ratios and horizontal flips Krizhevsky et al. (2012)). On CIFAR-10, we use horizontal flips and croppings of fixed size on a 2-padded input image. We use PyTorch version 1.0 for all our experiments. On ImageNet, we use the "Dali" library for fast data augmentation implementation [1].

- **ImageNet Labels - full dataset - AlexNet**: we train for 90 epochs with a total batch-size of 4096 distributed over 8 GPUs (512 samples per GPU), learning rate of 0.4, weight decay of 0.0001. We decay the learning rate by a factor 10 at epochs 30 and 60.

- **ImageNet Labels - full dataset - ResNet-50**: we train for 90 epochs with a total batch-size of 1536 distributed over 16 GPUs (96 samples per GPU), learning rate of 0.1, weight decay of 0.0001. We decay the learning rate by a factor 10 at epochs 50, 65 and 80.

- **ImageNet Labels - 30% dataset**: we train for 200 epochs with a total batch-size of 1536 distributed over 16 GPUs (96 samples per GPU), learning rate of 0.1 warmed up during the first 5 epochs, weight decay of 0.0003. We decay the learning rate by a factor 10 at epochs 140, 160 and 180.

- **ImageNet Labels - 30% of classes**: we train for 200 epochs with a total batch-size of 1536 distributed over 16 GPUs (96 samples per GPU), learning rate of 0.03 warmed up during the first 5 epochs, weight decay of 0.001. We decay the learning rate by a factor 10 at epochs 140, 160 and 180.

- **ImageNet Labels - 10% dataset**: we train for 200 epochs with a total batch-size of 768 distributed over 8 GPUs (96 samples per GPU), learning rate of 0.1 warmed up during the first 5 epochs, weight decay of 0.001. We decay the learning rate by a factor 10 at epochs 140, 160 and 180.

- **ImageNet Labels - 10% of classes**: we train for 200 epochs with a total batch-size of 768 distributed over 8 GPUs (96 samples per GPU), learning rate of 0.3 warmed up during the first 5 epochs, weight decay of 0.0003. We decay the learning rate by a factor 10 at epochs 140, 160 and 180.

- **ImageNet Labels - 1% dataset**: we train for 1000 epochs with a total batch-size of 768 distributed over 8 GPUs (96 samples per GPU), learning rate of 0.1, weight decay of 0.001. We decay the learning rate by a factor 10 at epochs 700, 800 and 900. We follow Zhai et al. (2019) and use random color data augmentation during training.

- **ImageNet Labels - 1% of classes**: we train for 1000 epochs with a total batch-size of 768 distributed over 8 GPUs (96 samples per GPU), learning rate of 0.01, weight decay of 0.003. We decay the learning rate by a factor 10 at epochs 700, 800 and 900. We follow Zhai et al. (2019) and use random color data augmentation during training.

- **ImageNet Semi-supervised RotNet**: We reproduce the semi-supervised method of Zhai et al. (2019) and follow precisely their hyperparameter. We train for 200 epochs with a total batch-size of 2048 distributed over 32 GPUs (64 samples per GPU), learning rate of 0.1 warmed up during the first 5 epochs, weight decay of 0.0003. We decay the learning rate by a factor 10 at epochs 140, 160 and 180.

- **ImageNet RotNet - ResNet-50**: we train for 90 epochs with a total batch-size of 1536 distributed over 16 GPUs (96 samples per GPU), learning rate of 1 warmed up during the first 5 epochs, weight decay of 0.00001. We decay the learning rate by a factor 10 at epochs 50, 65 and 80

- **ImageNet Exemplar - ResNet-50**: we train for 30 epochs with a total batch-size of 1536 distributed over 32 GPUs (48 samples per GPU), learning rate of 1, weight decay of 0.00001. We decay the learning rate by a factor 10 at epochs 20 and 25. We follow the

---

[1] github.com/NVIDIA/DALI

Exemplar implementation based on triplet margin loss from Doersch & Zisserman (2017). As the goal of the task is to learn invariance to data transformation, we use data color augmentation and random small rotations on top of standard data augmentation scheme.

- **ImageNet RotNet - AlexNet**: we train for 90 epochs with a total batch-size of 8192 distributed over 16 GPUs (512 samples per GPU), learning rate of 0.5 warmed up during the first 2 epochs, weight decay of 0.00001. We decay the learning rate by a factor 10 at epochs 30 and 60.

- **ImageNet Exemplar - AlexNet**: we train for 30 epochs with a total batch-size of 4096 distributed over 16 GPUs (256 samples per GPU), learning rate of 0.1, weight decay of 0.0001. We decay the learning rate by a factor 10 at epoch 20. We follow the Exemplar implementation based on triplet margin loss from Doersch & Zisserman (2017). We use data color augmentation and random small rotations on top of standard data augmentation scheme.

- **Places Labels - ResNet-50**: we train for 90 epochs with a total batch-size of 2048 distributed over 16 GPUs (128 samples per GPU), learning rate of 0.1, weight decay of 0.0001. We decay the learning rate by a factor 10 at epochs 50, 65 and 80.

- **CIFAR-10 Labels & RotNet - VGG-19 & ResNet-18**: we train for 160 epochs with a total batch-size of 512 on 1 GPU, learning rate of 0.001, weight decay of 0.0001. We decay the learning rate by a factor 10 at epochs 80 and 120.

- **CIFAR-10 Exemplar - VGG-19 & ResNet-18**: we train for 180 epochs with a total batch-size of 512 on 1 GPU, learning rate of 0.0003, weight decay of 0.0001. We decay the learning rate by a factor 10 at epochs 120. As the goal of the task is to learn invariance to data transformation, we use data color augmentation and random small rotations on top of standard data augmentation scheme.

- **CIFAR-10 Randomly permuted labels - VGG-19 & ResNet-18**: we train for 160 epochs with a total batch-size of 512 on 1 GPU, learning rate of 0.001, weight decay of 0.0001. We decay the learning rate by a factor 10 at epochs 80 and 120. We do not apply data augmentation.

## B  LATE RESETTING PARAMETER

We follow Frankle et al. (2019) and use late resetting (or *rewind*) for the winning tickets generation process. Indeed, before re-training a winning ticket, we reset its weights to their value "early in training" of the full over-parameterized network. In our work, we set the late resetting parameter to 1 epoch on CIFAR-10. However, when dataset size, total number of epochs, mini-batch sizes or learning rate vary, it becomes more complicated to choose a rewind criterion that guarantees a fair comparison between all settings. A choice can be to rewind at a point where "the same amount of information" has been processed. Thus, in our work, we choose to set the rewind parameter to $3 \times 1,280,000$ samples for all our experiments on ImageNet, which corresponds to 3 epochs on full ImageNet. We describe in Table 1 to what this rewind parameter corresponds to in terms of number of epoch, number of data samples seen, number of gradient update and percentage of total training for our different experiments. Moreover, we show in Figure 7 the performance of winning tickets generated using 10% of ImageNet with different values of rewind. Each of the considered value corresponds to keeping one of the criteria (number of epoch, number of data samples seen, number of gradient update or percentage of total training) fixed compared to the rewind parameter on full ImageNet (first row of Table 1).

| | # epochs | # samples seen | # gradient updates | % of training |
|---|---|---|---|---|
| ImageNet 100% | 3 | $\sim 3 \times 1,280,000$ | 2500 | 3.3% |
| ImageNet 30% | 10 | $\sim 3 \times 1,280,000$ | 2500 | 5.0% |
| ImageNet 10% | 30 | $\sim 3 \times 1,280,000$ | 5000 | 15% |
| ImageNet 1% | 300 | $\sim 3 \times 1,280,000$ | 5000 | 30% |

Table 1

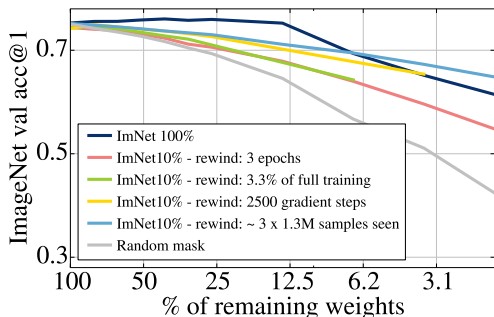

Figure 7: ImageNet top 1 validation accuracy of winning tickets generated with a subset of 10% of ImageNet dataset. We show the influence of different values for the late resetting parameter.

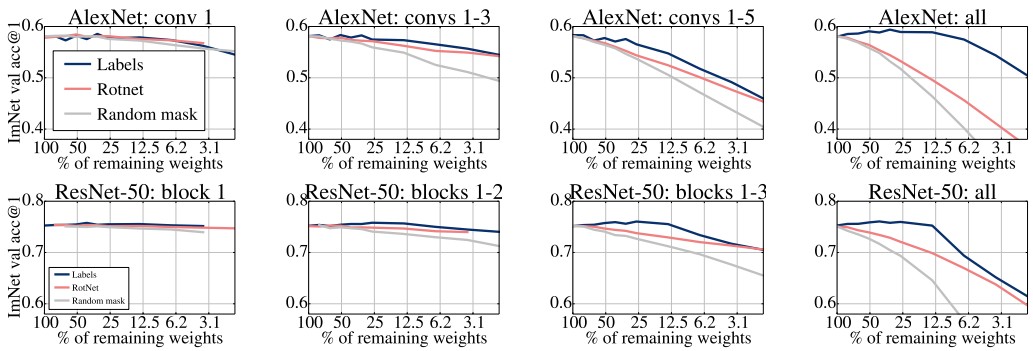

Figure 8: ImageNet top 1 validation accuracy of winning tickets generated by pruning partly or entirely (all) a network with 2 generation tasks: labels classification or RotNet. AlexNet is used in top row and ResNet-50 in the bottom row. We also show for reference results when the network layers are randomly pruned.

## C  LAYERWISE WINNING TICKETS

In this appendix we show more results on the layerwise winning tickets experiment (see 4.1). We generate winning tickets by pruning only the $n$ first convolutional layers of a network, and leaving the remaining of the network unpruned. On AlexNet (first row of Figure 8), we consider 4 situations. From left to right in the first row of Figure 8), we prune the first convolutiona layer; up to the third convolutiona layer; up to the fifth convolutional layer; or the whole network. For ResNet-50, from left to right in the bottom row of Figure 8), we prune the first 11 convolutiona layers; the first 24 convolutiona layers; the first 43 convolutional layers; or the whole network. Consistently with observations from the self-supervised literature, self-supervised winning tickets are competitive with labels ones when only the shallow and mid-level layers are pruned. Indeed, these layers are not specialized yet for the labels classification tasks and many works have shown that representations learned are similar at these depths.

## D  SPARSE TRAINED NETWORKS ON CIFAR-10

In this appendix, in Figure 9, we provide results on more architectures about the proportion of weights zeroed during training on CIFAR-10 compared to ImageNet. Note that we do not consider the batch-norm layers parameters, nor the parameters of the last fully-connected layer (since we do not prune it in our setting). For all the considered VGGs we use the modified version of Morcos et al. (2019), replacing the final MLP by a fully connected layer.

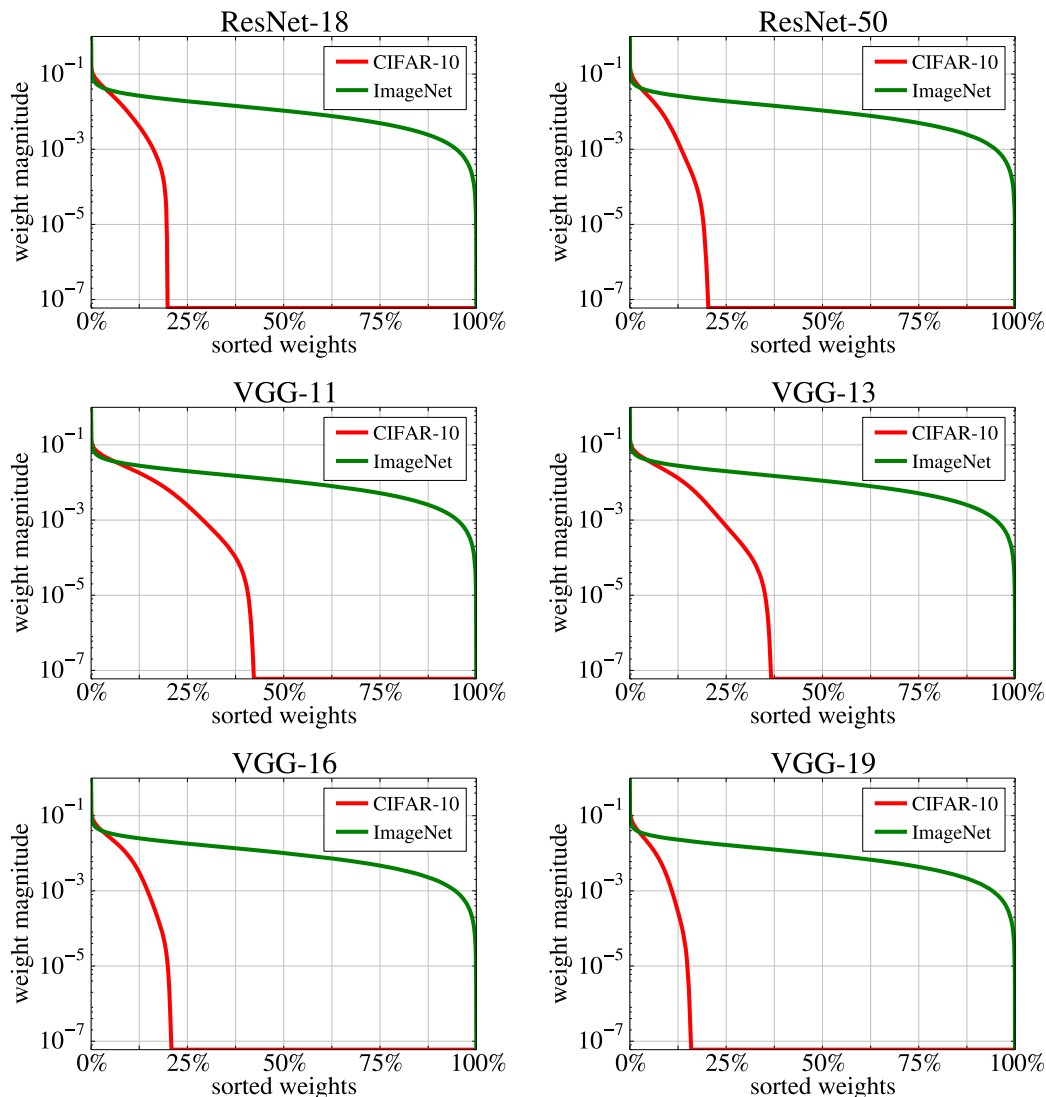

Figure 9: Magnitude of the weights of a trained network on two different datasets: CIFAR-10 (green) and ImageNet (red) with different datasets. We perform thresholding at machine precision value (bottom of y-axis).

## E  EXACT NUMBERS FROM FIGURES

In Tables 2 3 4 5 6 7 8, we report accuracies for each setting presented in the main paper. We report mean and standard errors for our experiments which we run with 3 different seeds for ImageNet and Places-365 and 5 different seeds for CIFAR-10.

## F  VARYING DATASET SIZE ON CIFAR-10

We choose to present results with ImageNet only in the main paper because varying the dataset size of an already small dataset such as CIFAR-10 is less conclusive. Moreover, as noted before 4.5, studying sparsity on CIFAR-10 should be done precautiously since trained deep networks are already sparse. In this Appendix, we present results on CIFAR-10 in Table 10. With VGG-19 architecture, we observe that decreasing the dataset size still leads to reasonable winning tickets. However, with ResNet-18 we observe an important performance gap of winning tickets generated with a fraction of

| Remaining weights | Labels | RotNet | Exemplar | Random | Random - adjusted |
|---|---|---|---|---|---|
| 1 | $86.54 \pm 0.08$ | | | | |
| 0.8 | $86.45 \pm 0.09$ | $86.66 \pm 0.11$ | $86.34 \pm 0.14$ | $86.05 \pm 0.10$ | |
| 0.64 | $86.45 \pm 0.06$ | $86.72 \pm 0.07$ | $86.36 \pm 0.06$ | $85.63 \pm 0.05$ | |
| 0.512 | $86.39 \pm 0.11$ | $86.61 \pm 0.06$ | $86.47 \pm 0.10$ | $85.22 \pm 0.11$ | |
| 0.410 | $86.55 \pm 0.06$ | $86.64 \pm 0.13$ | $86.64 \pm 0.06$ | $84.56 \pm 0.05$ | |
| 0.328 | $86.45 \pm 0.07$ | $86.55 \pm 0.06$ | $86.38 \pm 0.07$ | $83.73 \pm 0.05$ | |
| 0.262 | $86.48 \pm 0.13$ | $86.48 \pm 0.10$ | $86.41 \pm 0.10$ | $83.05 \pm 0.09$ | |
| 0.134 | $86.47 \pm 0.05$ | $86.53 \pm 0.06$ | $85.34 \pm 0.05$ | $80.97 \pm 0.13$ | $85.65 \pm 0.12$ |
| 0.069 | $86.96 \pm 0.06$ | $86.44 \pm 0.15$ | $84.86 \pm 0.03$ | $78.68 \pm 0.10$ | $83.73 \pm 0.08$ |
| 0.035 | $86.91 \pm 0.08$ | $85.90 \pm 0.05$ | $84.40 \pm 0.08$ | $76.03 \pm 0.19$ | $81.97 \pm 0.14$ |
| 0.018 | $86.80 \pm 0.07$ | $85.54 \pm 0.10$ | $84.07 \pm 0.14$ | $72.11 \pm 0.29$ | $79.66 \pm 0.10$ |
| 0.009 | $86.41 \pm 0.06$ | $84.59 \pm 0.09$ | $82.71 \pm 0.12$ | $65.90 \pm 0.30$ | $77.04 \pm 0.27$ |
| 0.005 | $85.76 \pm 0.13$ | $83.43 \pm 0.11$ | $79.69 \pm 0.11$ | $59.71 \pm 0.43$ | $72.34 \pm 0.28$ |
| 0.002 | $84.50 \pm 0.05$ | $80.45 \pm 0.09$ | $75.59 \pm 0.16$ | $52.08 \pm 0.35$ | $66.67 \pm 0.38$ |
| 0.001 | $81.71 \pm 0.12$ | $75.41 \pm 0.18$ | $70.60 \pm 0.24$ | $34.29 \pm 2.59$ | $60.31 \pm 0.29$ |

Table 2: Means and standard errors for accuracies from Figure 1b (*Label-agnostic winning tickets: CIFAR*-10 *ResNet*-18).

| Remaining weights | Labels | RotNet | Exemplar | Random | Random - adjusted |
|---|---|---|---|---|---|
| 1 | $92.62 \pm 0.05$ | | | | |
| 0.8 | $92.45 \pm 0.05$ | $92.48 \pm 0.08$ | $92.28 \pm 0.06$ | $92.26 \pm 0.06$ | |
| 0.64 | $92.58 \pm 0.08$ | $92.31 \pm 0.07$ | $92.29 \pm 0.06$ | $91.93 \pm 0.05$ | |
| 0.512 | $92.46 \pm 0.05$ | $92.38 \pm 0.04$ | $92.17 \pm 0.07$ | $91.82 \pm 0.05$ | |
| 0.410 | $92.44 \pm 0.07$ | $92.30 \pm 0.07$ | $92.33 \pm 0.09$ | $91.58 \pm 0.04$ | |
| 0.328 | $92.51 \pm 0.06$ | $92.63 \pm 0.07$ | $92.39 \pm 0.05$ | $91.24 \pm 0.08$ | |
| 0.262 | $92.56 \pm 0.04$ | $92.76 \pm 0.09$ | $92.51 \pm 0.05$ | $90.76 \pm 0.09$ | |
| 0.134 | $92.61 \pm 0.06$ | $92.67 \pm 0.04$ | $92.57 \pm 0.06$ | $89.61 \pm 0.05$ | $92.41 \pm 0.05$ |
| 0.069 | $92.68 \pm 0.08$ | $92.68 \pm 0.07$ | $92.31 \pm 0.07$ | $87.86 \pm 0.14$ | $91.76 \pm 0.06$ |
| 0.035 | $92.64 \pm 0.10$ | $92.17 \pm 0.08$ | $91.83 \pm 0.05$ | $85.62 \pm 0.11$ | $90.54 \pm 0.09$ |
| 0.018 | $92.40 \pm 0.06$ | $91.57 \pm 0.08$ | $90.90 \pm 0.04$ | $82.49 \pm 0.15$ | $88.67 \pm 0.04$ |
| 0.009 | $92.30 \pm 0.07$ | $90.45 \pm 0.06$ | $89.53 \pm 0.08$ | $78.26 \pm 0.38$ | $86.36 \pm 0.11$ |
| 0.005 | $91.89 \pm 0.08$ | $89.22 \pm 0.10$ | $88.21 \pm 0.07$ | $70.68 \pm 0.47$ | $83.62 \pm 0.12$ |
| 0.002 | $90.76 \pm 0.08$ | $87.27 \pm 0.17$ | $85.69 \pm 0.08$ | $59.21 \pm 1.19$ | $79.30 \pm 0.30$ |
| 0.001 | $76.81 \pm 4.61$ | $81.63 \pm 0.58$ | $77.04 \pm 0.30$ | $27.18 \pm 7.04$ | $71.29 \pm 0.44$ |

Table 3: Means and standard errors for accuracies from Figure 1b (*Label-agnostic winning tickets: CIFAR*-10 *VGG*-19).

the dataset compared to using the entire dataset. Indeed, when only only 10% of CIFAR-10 (5000 images) the generated winning tickets are at random level.

## G    RANDOM RE-INITIALIZATION OF WINNING TICKETS

In this Appendix, we present results with random re-initialization for winning tickets found with labels or with RotNet self-supervised task on both ImageNet and CIFAR-10 in Figure 11. On ImageNet 11b, consistently with the experiments of Frankle et al. (2019), we observe that resetting the weights accordingly is crucial to get high accuracy. Indeed, on both ResNet-50 and AlexNet, for labels, the subnetworks that are reset to their weights early in training (dark blue plain line) perform significantly better than subnetworks randomly re-initialized (dark blue dashed line). Interestingly, this is not the case for RotNet winning tickets: starting from original weights (pink plain line) gives only a very slight boost (or even no boost at all) of performance compared to randomly re-initialization (pink dashed line). Overall, labels or RotNet subnetworks perform in the same ball park when randomly re-initialized, but using the original weights gives a large boost of performance for labels but not for RotNet. Thus, it suggests that the information carried by the pruned mask itself

| Remaining weights | Labels | RotNet | Exemplar | Random |
|---|---|---|---|---|
| 1 | $75.26 \pm 0.04$ | | | |
| 0.8 | $75.57 \pm 0.05$ | $74.92 \pm 0.02$ | $73.53 \pm 0.07$ | $74.22 \pm 0.27$ |
| 0.64 | $75.58 \pm 0.05$ | $74.33 \pm 0.06$ | $73.03 \pm 0.04$ | $73.52 \pm 0.44$ |
| 0.512 | $75.86 \pm 0.01$ | $73.92 \pm 0.04$ | $72.44 \pm 0.07$ | $72.71 \pm 0.54$ |
| 0.410 | $76.06 \pm 0.02$ | $73.42 \pm 0.02$ | $71.92 \pm 0.03$ | $71.68 \pm 0.59$ |
| 0.328 | $75.79 \pm 0.01$ | $72.87 \pm 0.04$ | $71.23 \pm 0.06$ | $70.46 \pm 0.87$ |
| 0.262 | $75.96 \pm 0.01$ | $72.08 \pm 0.09$ | $70.68 \pm 0.00$ | $69.34 \pm 1.02$ |
| 0.134 | $75.24 \pm 0.07$ | $69.88 \pm 0.05$ | $68.15 \pm 0.10$ | $64.56 \pm 1.82$ |
| 0.069 | $69.38 \pm 0.10$ | $66.94 \pm 0.03$ | $65.30 \pm 0.09$ | $56.71 \pm 3.73$ |
| 0.035 | $65.16 \pm 0.14$ | $63.77 \pm 0.16$ | $61.96 \pm 0.10$ | $51.09 \pm 4.41$ |
| 0.018 | $61.45 \pm 0.23$ | $59.68 \pm 0.06$ | $57.61 \pm 0.19$ | $42.40 \pm 5.70$ |
| 0.009 | $57.19 \pm 0.27$ | $54.52 \pm 0.22$ | $51.75 \pm 0.12$ | $33.10 \pm 6.45$ |
| 0.005 | $52.14 \pm 0.21$ | $47.89 \pm 0.13$ | $45.44 \pm 0.05$ | $23.17 \pm 5.86$ |
| 0.002 | $43.77 \pm 0.15$ | $39.07 \pm 0.37$ | $38.06 \pm 0.13$ | $9.75 \pm 3.52$ |
| 0.001 | $32.17 \pm 0.48$ | $28.71 \pm 0.38$ | $29.70 \pm 0.11$ | $0.62 \pm 0.42$ |

Table 4: Means and standard errors for accuracies from Figure 1b (*Label-agnostic winning tickets: ImageNet ResNet*-50).

| Remaining weights | Labels | RotNet | Exemplar | Random |
|---|---|---|---|---|
| 1 | $58.01 \pm 0.07$ | | | |
| 0.8 | $58.54 \pm 0.25$ | $57.80 \pm 0.02$ | $56.21 \pm 0.10$ | $57.61 \pm 0.06$ |
| 0.64 | $58.62 \pm 0.17$ | $57.04 \pm 0.12$ | $55.51 \pm 0.08$ | $56.86 \pm 0.11$ |
| 0.512 | $59.06 \pm 0.21$ | $56.42 \pm 0.07$ | $54.48 \pm 0.03$ | $55.82 \pm 0.14$ |
| 0.410 | $58.83 \pm 0.21$ | $55.37 \pm 0.10$ | $53.51 \pm 0.11$ | $54.83 \pm 0.02$ |
| 0.328 | $59.39 \pm 0.13$ | $54.41 \pm 0.07$ | $52.36 \pm 0.12$ | $53.20 \pm 0.08$ |
| 0.262 | $58.93 \pm 0.20$ | $53.26 \pm 0.05$ | $50.99 \pm 0.08$ | $51.75 \pm 0.02$ |
| 0.134 | $58.87 \pm 0.09$ | $49.58 \pm 0.07$ | $46.63 \pm 0.01$ | $46.36 \pm 0.07$ |
| 0.069 | $57.46 \pm 0.08$ | $45.62 \pm 0.07$ | $41.53 \pm 0.12$ | $40.18 \pm 0.06$ |
| 0.035 | $54.36 \pm 0.22$ | $41.09 \pm 0.20$ | $36.61 \pm 0.17$ | $33.47 \pm 0.05$ |
| 0.018 | $50.43 \pm 0.19$ | $36.54 \pm 0.16$ | $31.23 \pm 0.16$ | $26.45 \pm 0.09$ |
| 0.009 | $45.22 \pm 0.07$ | $31.46 \pm 0.09$ | $27.03 \pm 0.06$ | $20.91 \pm 0.34$ |
| 0.005 | $37.42 \pm 0.14$ | $26.20 \pm 0.11$ | $23.38 \pm 0.14$ | $16.28 \pm 0.21$ |
| 0.002 | $28.66 \pm 0.31$ | $20.66 \pm 0.18$ | $19.34 \pm 0.20$ | $11.29 \pm 0.22$ |
| 0.001 | $21.18 \pm 0.24$ | $15.46 \pm 0.12$ | $15.53 \pm 0.03$ | $1.73 \pm 1.33$ |

Table 5: Means and standard errors for accuracies from Figure 1b (*Label-agnostic winning tickets: ImageNet AlexNet*).

is similar for labels and RotNet subnetworks but the weights of the RotNet winning tickets are not as good starting points as the weights from labels winning tickets.

On CIFAR-10 11a, up to a certain level of sparsity (that roughly corresponds to the natural level of sparsity of the network), using random re-initialization or weights "early in training" gives similar performance. However, for more extreme pruning rates, using a particular set of weights gives significantly better performance than random re-initialization. Our results are consistent with the findings of Liu et al. (2019) (Figure 7.a of their paper), Frankle et al. (2019) (Appendix A of their paper) Morcos et al. (2019) (Appendix A2 of their paper).

| Remaining weights | 100% ImNet | 30% ImNet | 10% ImNet | 1% ImNet | Random |
|---|---|---|---|---|---|
| 1 | $75.26 \pm 0.04$ | | | | |
| 0.8 | $75.57 \pm 0.05$ | $74.58 \pm 0.07$ | $74.86 \pm 0.03$ | $73.86 \pm 0.07$ | $74.22 \pm 0.27$ |
| 0.64 | $75.58 \pm 0.05$ | $74.49 \pm 0.04$ | $74.52 \pm 0.04$ | $73.28 \pm 0.04$ | $73.52 \pm 0.44$ |
| 0.512 | $75.86 \pm 0.01$ | $74.38 \pm 0.06$ | $74.15 \pm 0.05$ | $72.66 \pm 0.07$ | $72.71 \pm 0.54$ |
| 0.410 | $76.06 \pm 0.02$ | $74.28 \pm 0.01$ | $73.73 \pm 0.02$ | $71.95 \pm 0.06$ | $71.68 \pm 0.59$ |
| 0.328 | $75.79 \pm 0.01$ | $73.95 \pm 0.04$ | $73.37 \pm 0.04$ | $71.20 \pm 0.06$ | $70.46 \pm 0.87$ |
| 0.262 | $75.96 \pm 0.01$ | $73.32 \pm 0.10$ | $73.01 \pm 0.09$ | $70.47 \pm 0.03$ | $69.34 \pm 1.02$ |
| 0.134 | $75.24 \pm 0.07$ | $70.21 \pm 0.03$ | $71.15 \pm 0.06$ | $67.70 \pm 0.05$ | $64.56 \pm 1.82$ |
| 0.069 | $69.38 \pm 0.10$ | $67.10 \pm 0.08$ | $69.48 \pm 0.04$ | $64.49 \pm 0.20$ | $56.71 \pm 3.73$ |
| 0.035 | $65.16 \pm 0.14$ | $64.01 \pm 0.04$ | $67.28 \pm 0.04$ | $60.61 \pm 0.37$ | $51.09 \pm 4.41$ |
| 0.018 | $61.45 \pm 0.23$ | $60.77 \pm 0.08$ | $64.84 \pm 0.08$ | $55.48 \pm 0.95$ | $42.40 \pm 5.70$ |
| 0.009 | $57.19 \pm 0.27$ | $56.76 \pm 0.06$ | $61.55 \pm 0.12$ | $46.43 \pm 2.70$ | $33.10 \pm 6.45$ |
| 0.005 | $52.14 \pm 0.21$ | $52.03 \pm 0.10$ | $55.56 \pm 0.30$ | $25.61 \pm 10.52$ | $23.17 \pm 5.86$ |
| 0.002 | $43.77 \pm 0.15$ | $44.50 \pm 0.46$ | $46.89 \pm 0.27$ | $18.53 \pm 5.37$ | $9.75 \pm 3.52$ |
| 0.001 | $32.17 \pm 0.48$ | $32.76 \pm 0.35$ | $36.62 \pm 0.10$ | $12.06 \pm 3.27$ | $0.62 \pm 0.42$ |

Table 6: Means and standard errors for accuracies from Figure 3a (*Varying dataset size*).

| Remaining weights | Full ImNet | 30% ImNet classes | 10% ImNet classes | 1% ImNet classes | Random |
|---|---|---|---|---|---|
| 1 | $75.26 \pm 0.04$ | | | | |
| 0.8 | $75.57 \pm 0.05$ | $74.53 \pm 0.03$ | $74.63 \pm 0.05$ | $74.44 \pm 0.01$ | $74.22 \pm 0.27$ |
| 0.64 | $75.58 \pm 0.05$ | $74.20 \pm 0.02$ | $74.08 \pm 0.10$ | $73.79 \pm 0.05$ | $73.52 \pm 0.44$ |
| 0.512 | $75.86 \pm 0.01$ | $73.70 \pm 0.05$ | $73.75 \pm 0.05$ | $73.33 \pm 0.05$ | $72.71 \pm 0.54$ |
| 0.410 | $76.06 \pm 0.02$ | $72.95 \pm 0.04$ | $73.30 \pm 0.05$ | $72.51 \pm 0.10$ | $71.68 \pm 0.59$ |
| 0.328 | $75.79 \pm 0.01$ | $72.07 \pm 0.06$ | $72.86 \pm 0.02$ | $71.85 \pm 0.11$ | $70.46 \pm 0.87$ |
| 0.262 | $75.96 \pm 0.01$ | $71.22 \pm 0.05$ | $72.23 \pm 0.03$ | $71.05 \pm 0.07$ | $69.34 \pm 1.02$ |
| 0.134 | $75.24 \pm 0.07$ | $68.27 \pm 0.08$ | $70.28 \pm 0.09$ | $68.25 \pm 0.11$ | $64.56 \pm 1.82$ |
| 0.069 | $69.38 \pm 0.10$ | $65.10 \pm 0.08$ | $67.99 \pm 0.10$ | $64.94 \pm 0.07$ | $56.71 \pm 3.73$ |
| 0.035 | $65.16 \pm 0.14$ | $61.70 \pm 0.04$ | $65.01 \pm 0.04$ | $60.92 \pm 0.14$ | $51.09 \pm 4.41$ |
| 0.018 | $61.45 \pm 0.23$ | $58.39 \pm 0.09$ | $61.78 \pm 0.07$ | $55.66 \pm 0.14$ | $42.40 \pm 5.70$ |
| 0.009 | $57.19 \pm 0.27$ | $54.45 \pm 0.08$ | $57.20 \pm 0.06$ | $49.78 \pm 0.19$ | $33.10 \pm 6.45$ |
| 0.005 | $52.14 \pm 0.21$ | $49.38 \pm 0.06$ | $51.66 \pm 0.26$ | $42.65 \pm 0.23$ | $23.17 \pm 5.86$ |
| 0.002 | $43.77 \pm 0.15$ | $42.57 \pm 0.09$ | $45.12 \pm 0.30$ | $34.56 \pm 0.19$ | $9.75 \pm 3.52$ |
| 0.001 | $32.17 \pm 0.48$ | $30.10 \pm 0.32$ | $37.36 \pm 0.03$ | $27.54 \pm 0.08$ | $0.62 \pm 0.42$ |

Table 7: Means and standard errors for accuracies from Figure 3b (*Varying number of classes*).

| Remaining weights | Semi-Sup | Places-365 | CIFAR-10 |
|---|---|---|---|
| 0.8 | $74.89 \pm 0.06$ | $74.27 \pm 0.07$ | $72.57 \pm 0.22$ |
| 0.64 | $74.76 \pm 0.04$ | $73.90 \pm 0.02$ | $72.13 \pm 0.16$ |
| 0.512 | $74.60 \pm 0.07$ | $73.48 \pm 0.03$ | $71.58 \pm 0.26$ |
| 0.410 | $74.48 \pm 0.08$ | $73.19 \pm 0.08$ | $70.78 \pm 0.28$ |
| 0.328 | $74.22 \pm 0.03$ | $72.61 \pm 0.03$ | $69.77 \pm 0.21$ |
| 0.262 | $74.03 \pm 0.02$ | $71.91 \pm 0.08$ | $68.76 \pm 0.27$ |
| 0.134 | $72.59 \pm 0.08$ | $69.29 \pm 0.11$ | $65.65 \pm 0.29$ |
| 0.069 | $70.09 \pm 0.05$ | $65.93 \pm 0.09$ | $61.61 \pm 0.35$ |
| 0.035 | $67.15 \pm 0.08$ | $61.62 \pm 0.10$ | $56.03 \pm 0.46$ |
| 0.018 | $62.90 \pm 0.15$ | $57.24 \pm 0.13$ | $47.53 \pm 0.47$ |
| 0.009 | $58.40 \pm 0.11$ | $52.95 \pm 0.10$ | $36.95 \pm 0.71$ |

Table 8: Means and standard errors for accuracies from Figure 5 (*(a) Semi-supervised & (b) Transfer between datasets*).

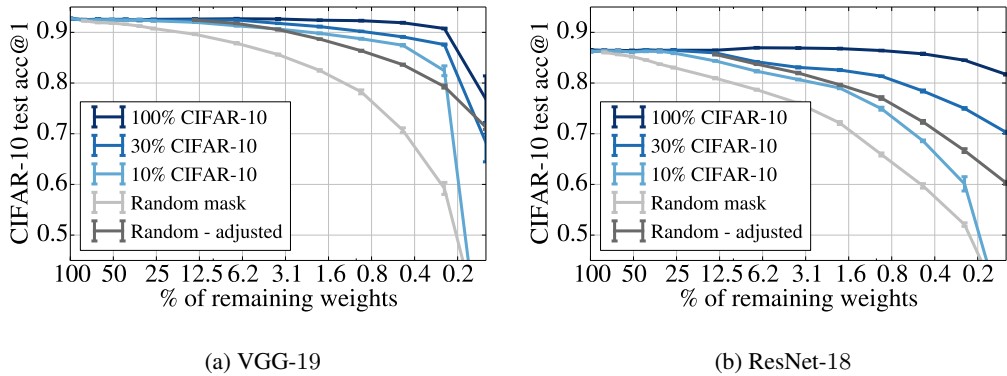

(a) VGG-19      (b) ResNet-18

Figure 10: Top-1 accuracy on CIFAR-10 test set of winning tickets generated with different training data. The x-axis represents different levels of winning ticket sparsity.

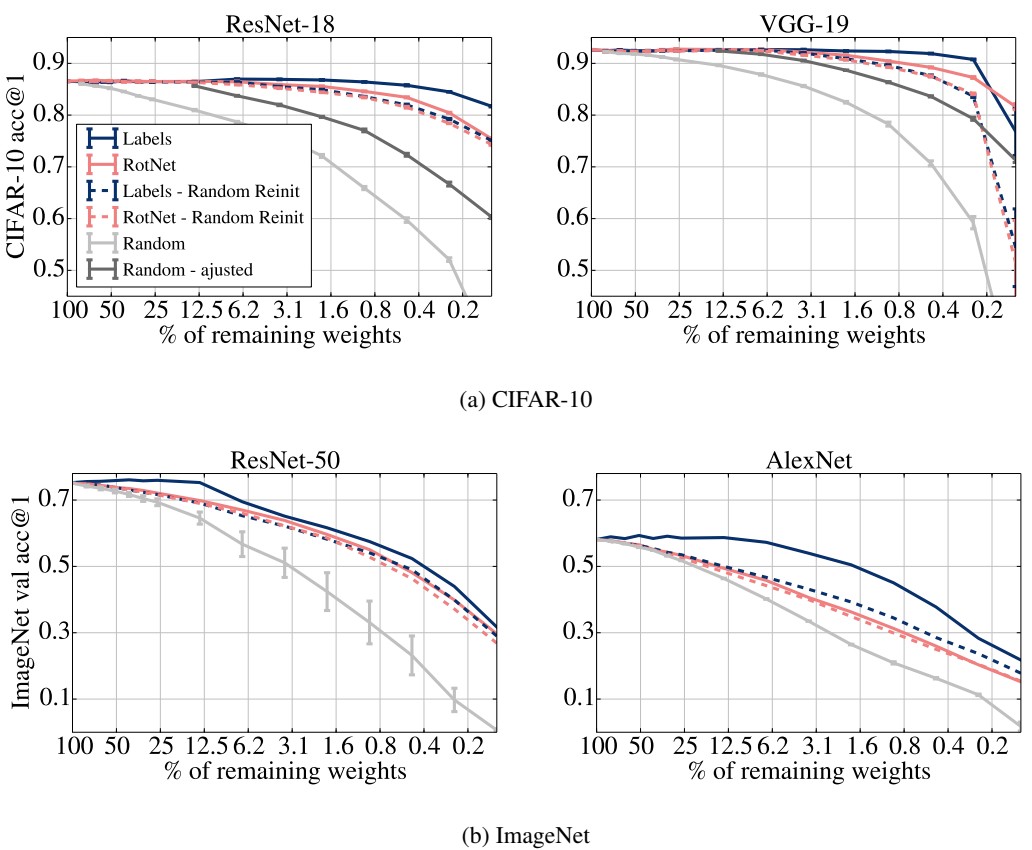

(a) CIFAR-10

(b) ImageNet

Figure 11: We report CIFAR-10 test (b) and ImageNet val (a) top-1 accuracy for winning tickets found with RotNet label-agnostic task. The x-axis corresponds to different pruning ratios. We compare the performance with supervised winning tickets and random subnetworks (randomly drawn weights from the initialization distribution and randomly permuted masks). On CIFAR-10, deep models are highly sparse with only $\sim 15\%$ of non-zero weights. Thus, we adjust the random baseline to start with the correct mask at the natural level of network sparsity (4.5 for details). The dashed lines correspondong to applying the pruning masks found with labels or RotNet, but with randomly re-initialized weights.

