# OpenReview forum: "Finding Winning Tickets with Limited (or No) Supervision"
_ICLR.cc/2020/Conference — Reject_

### Official Review · AnonReviewer3 · 2019-10-23
**Official Blind Review #3**

**Rating:** 1

**Review:**

This paper studies the problem of finding sparse networks in a limited supervision setup. The authors build on the lottery ticket work of Frankle & Carbin and investigate the validity of their idea when one has few or no labels. This work is an immediate followup on Morcos et al. who investigated the transferability of lottery tickets.

This work is more observational rather than algorithmic or theoretical. Authors study various small sample/label setups where network sparsification works well.

Main contribution is Section 4.1 where self-supervision is investigated. However given that lottery tickets are transferable (Morcos paper) it is really not that surprising that semisupervised learning algorithms will do a decent job as well. I also don't see a practical benefit beyond transfer learning setup.

Section 4.2 essentially sweeps through supervised problem parameters such as reducing sample size, adding noise etc and . The main application seems to be extracting lottery tickets faster by downsampling the data however this aspect is again fairly obvious.

In short, unfortunately, this paper doesn't cut it for ICLR. As improvements, I would recommend adding standard semi-supervised training techniques to their comparison. I was surprised to not see pseudo-labeling or consistency training (e.g. virtual adversarial training).

**Experience Assessment:**

I have published in this field for several years.

**Review Assessment: Checking Correctness Of Derivations And Theory:**

N/A

**Review Assessment: Checking Correctness Of Experiments:**

I assessed the sensibility of the experiments.

**Review Assessment: Thoroughness In Paper Reading:**

I read the paper at least twice and used my best judgement in assessing the paper.

---

> ### Author Response · Authors · 2019-11-15
> **Reply to review #3**
>
> We agree with the reviewer that our contribution is not methodological but mainly experimental.
>
> Also, we agree that some of our observations might be intuitively expected and reasonable, though we note that the fact that winning tickets transfer between similar datasets with labels (as shown in Morcos et al., 2019) does not necessarily imply that transfer from self-supervised tasks should be possible. Based on previous results, it is entirely plausible that winning tickets are dependent on labels (i.e., p(y|x) vs. p(x)).
> Furthermore, we argue that, even if these results are expected, confirming these intuitions with rigorous experimentations as we propose in our paper is still important, as noted by R2 (“kind of expected, but it is still good that this paper provide solid experimental results to verify this”).
>
> “I also don't see a practical benefit beyond transfer learning setup.“: As a byproduct, the label-agnostic winning tickets also allow to study the transferability of winning tickets between different tasks, which has a concrete practical benefit. Indeed, similarly to the motivation of Morcos et al., if winning tickets can transfer between tasks, then they can be reused across a variety of problems, thus dispensing the need for generating new winning tickets for each new task.
>
> “given that lottery tickets are transferable (Morcos paper) it is really not that surprising”: The datasets used in the paper of Morcos et al. were pretty similar to one another, and the fact that we can transfer between similar supervised tasks does not suggest that we should be able to transfer from self-supervised to supervised. Our work provides insights regarding the dependence of winning tickets on p(x) vs. p(y|x).
>
> "I was surprised to not see pseudo-labeling or consistency training": For our semi-supervised experiment, we choose to focus solely on the semi-supervised technique introduced in the paper "S4L: Self-Supervised Semi-Supervised Learning" of Zhai et al. (ICCV 2019) because it yields better performance compared to VAT or pseudo-labeling on ImageNet (see Table 1. from their paper).

---

### Official Review · AnonReviewer1 · 2019-10-24
**Official Blind Review #1**

**Rating:** 3

**Review:**

In the original lottery ticket paper, it points out that training the pruned architecture from scratch with initial weight can achieve the same performance compared to fine-tuning it. This work further discuss this phenomenon when data or label is not enough. It is good to see the few-data/label can still provide a comparable results. But some experiment’s results and its setting are confusing, while also makes me concerned about the conclusion solidness.

1) Usually in classification task (especially in cifar10 dataset), 0.5% to 1% accuracy could be a huge gap between two models. For example, in the original “Lottery Ticket Hypothesis” paper, using initial weight only has roughly 0.5% improvement compared to random initialization. But the figures in this paper do not contain a zoom-in details for each line, make me hard to distinguish the performance between each setting. If the author does not provide a detailed version, it will look like theses model have the same performance, which is actually wrong. The author should either plot a zoom-in figure especially when the pruning ratio larger than 50% or give a Table with accuracy of each setting. And it is better to complete the figure with several random seed and plot the error bar to avoid randomness.

2) Does the “Random - adjusted” item in Figure. 1 mean the correctly pruning architecture with random initialization? In "rethinking the value of network pruning", Liu et al. points that in the large learning rate setting (lr=0.1, which is also your setting), random initialization can achieve the same performance compared to the lottery ticket. In my perspective, I want to see whether few-data/label also works on random initialization instead of lottery tickets. I expect the author to explain the “Random -adjusted” experiment setting clearly in the response and I suggest the author to discuss the
random initialization part specifically.

3) Figure.3 only shows the “varying dataset size” experiments on ImageNet. The experiments on cifar10 is lacked. The author should complete this part in the response.

**Experience Assessment:**

I have published one or two papers in this area.

**Review Assessment: Checking Correctness Of Derivations And Theory:**

I carefully checked the derivations and theory.

**Review Assessment: Checking Correctness Of Experiments:**

I carefully checked the experiments.

**Review Assessment: Thoroughness In Paper Reading:**

I read the paper at least twice and used my best judgement in assessing the paper.

---

> ### Author Response · Authors · 2019-11-15
> **Reply to review #1**
>
> 1) Following the reviewer's comment, we report in Appendix E tables the exact accuracies in each setting. We report mean and standard errors for our experiments which we run with 3 (ImageNet and Places) or 6 (CIFAR) different seeds. We thank the reviewer for this recommendation and for helping us improving the clarity and robustness of the comparison.
>
> 2) The “Random - adjusted” baseline is not obtained by applying the pruning mask to randomly initialized weights. In the following lines, we motivate and clarify this baseline and have included this explanation in the paper updated version.
> We find that deep architectures (VGG-19 or ResNet-18 for example) trained on CIFAR-10 are naturally sparse (~80% of the weights are zeroed at convergence). Pruning a network at rates below its level of natural sparsity without impacting the performance is trivial because the network is already sparse. Indeed, we found that in the random global pruning baseline (which can remove non zero weights), pruning at rates below the natural sparsity of the network degrades accuracy, while pruning of weights that are already zeroed has no effect. Experiments performed with pruning rates below the natural level of sparsity of the network (~80%) are uninformative. Inconveniently, this performance gap carries over to higher pruning rates (in which we are interested in) and can lead to misleading interpretations. The random adjusted baseline removes this effect by first pruning the weights that naturally converge to zero after training. Then, we randomly mask the remaining non-zeroed weights to get different final desired pruning rates. The remaining non-masked weights are randomly initialized. This baseline therefore corrects for the natural sparsity present in CIFAR-10 networks.
>
> Regarding the random initialization remark, Liu et al. indeed show in Figure 7.a (unstructured iterative pruning) that starting from randomly reinitialized weights works well on deep architectures (VGG-16 and ResNet-50) on CIFAR-10 when pruned at rates below ~90%. This is consistent with the observations of Frankle et al. (2019) in the Appendix A of their paper. Indeed, Frankle et al. (2019) also show that up to a certain level of sparsity, training the subnetwork from its original weights or from random re-initialization gives comparable performance. However, for more extreme pruning rates (>90%), resetting the subnetwork to its original weights gives better performance than random re-initialization. In this work, we follow up on the work of Frankle et al. (2019) and Morcos et al. (2019) that both provide empirical evidence that in the regime of large datasets or high pruning rates, starting from a particular set of weights instead of random initialization is critical to reach high accuracy.
> For completeness though, we take into account the remark of the reviewer and have included in Appendix G results with random re-initialization for winning tickets found with labels or with RotNet self-supervised task on both ImageNet and CIFAR-10.
> On ImageNet, consistently with the experiments of Frankle et al. (2019) we observe that resetting the weights accordingly is crucial to get high accuracy. Indeed, on both ResNet-50 and AlexNet, for labels, the subnetworks that are reset to their weights early in training (dark blue plain line) perform significantly better than subnetworks randomly re-initialized (dark blue dashed line). Interestingly, this is not the case for RotNet winning tickets: starting from original weights (pink plain line) gives only a very slight boost (or even no boost at all) of performance compared to randomly re-initialization (pink dashed line). Overall, labels or rotnet subnetworks perform in the same ball park when randomly re-initialized, but using the original weights gives a large boost of performance for labels but not for rotnet. Thus, it suggests that the information carried by the pruned mask itself is similar for labels and rotnet subnetworks but the weights of the rotnet winning tickets are not as good starting points as the weights from labels winning tickets. We thank the reviewer for suggesting this experiment; it gives interesting insights on the difference of performance between labels and rotnet winning tickets.
> On CIFAR-10, up to a certain level of sparsity that roughly corresponds to the natural level of sparsity of the network, using random re-initialization or weights ‘early in training’ gives similar performance. However, for more extreme pruning rates, using a particular set of weights gives significantly better performance than random re-initialization.
>
> 3) We chose not to vary the dataset size on CIFAR-10 because it is already small. However, following the reviewer recommendation we include results with CIFAR-10 in Appendix F.
>
> Overall, we hope that our updated version of the paper along with our comments provide clarifications about our experimental settings and reinforce the validity of our results.

---

### Official Review · AnonReviewer4 · 2019-11-01
**Official Blind Review #4**

**Rating:** 6

**Review:**

In this paper, the authors try to provide empirical answers to several important open questions on winning tickets. They conduct most of experiments on ImageNet and results show that winning ticket is robust, and few data samples can also obtain good winning tickets.

Generally, the paper has conducted extensive experiments on three open questions and results prove their assumptions.

As describe in page 7, lottery tickets are sensitive to data distributions. I’m wondering, whether there will be winning ticket for multi-task learning with limited data each task? Will this be helpful in distilling the model?

**Experience Assessment:**

I do not know much about this area.

**Review Assessment: Checking Correctness Of Derivations And Theory:**

I assessed the sensibility of the derivations and theory.

**Review Assessment: Checking Correctness Of Experiments:**

I assessed the sensibility of the experiments.

**Review Assessment: Thoroughness In Paper Reading:**

I made a quick assessment of this paper.

---

> ### Author Response · Authors · 2019-11-15
> **Reply to review #4**
>
> We thank the reviewer for this positive feedback. We did not experiment on the particular scenario of multi-task learning with limited amount of data; we agree that this is an interesting problem for future work.

---

### Official Review · AnonReviewer2 · 2019-11-04
**Official Blind Review #2**

**Rating:** 3

**Review:**

This paper empirically studies the lottery ticket hypothesis with limited or no supervision. First, the authors use self-supervised learning to generate winning tickets, showing that "good" (reasonable) winning tickets can be found without labels. Second, the authors show that finding "good" (reasonable) winning tickets can be accelerated by a factor 5 on ImageNet by using only a subset of the data. The authors also argue that using large datasets is important to study lottery tickets, since deep networks trained on CIFAR-10 are natually sparse, making conclusions potentially misleading.

The experimental results are rich and provide more understanding of winning ticket generation with limited or no supervision. The results on self-supervised learning task (including the layer-wise pruning results) and a subset of training dataset are reasonable and kind of expected, but it is still good that this paper provide solid experimental results to verify this. As the paper observed, "none of the tickets found with limited access to labels and or data matches the accuracy of tickets found with all the labeled data when considering moderate pruning rates (more than 10% of unpruned weights)
on ImageNet. Indeed, we consistently observe a decrease in performance compared to the full overparametrized network as soon as we prune the network." In this sense, winning tickets are certainly label and data dependant. This undermines the *bold* claim in the abstract that "we provide a positive answer to both questions, by generating winning tickets with limited access to data, or with self-supervision". From my perspective, the ability to exactly perserve the accuracy while pruning the weights (see the flat regions of "Lables" curves in Figure 1,2,3,4,5) is the interesting part of the lottery ticket hypothesis. We have several different ways to achieve a descreased accuracy with a smaller network, the dynamics there may be a mixture of the lottery ticket hypothesis and standard model pruning, which needs more careful experiment design to separate different dynamics.

"using large datasets is important to study lottery tickets, since deep networks trained on CIFAR-10 are natually sparse, making conclusions potentially misleading." "The definition of “early in training” is somehow ill-defined: network
weights change much more for the first epochs than for the last ones." These two messages are important to future study of the lottery ticket hypothesis. This paper raises the issue of ill-definedness of “early in training”, but did not provide a solution.

Overall, I found that the experimental results in this paper are solid and provide more understandings of the lottery ticket hypothesis. However, I feel that the novelty of this paper is limited, and do not provide much new insights. Therefore, it does not reach the bar of being published at ICLR, from my perspective. Therefore, I say "Weak Reject".

**Experience Assessment:**

I do not know much about this area.

**Review Assessment: Checking Correctness Of Derivations And Theory:**

N/A

**Review Assessment: Checking Correctness Of Experiments:**

I assessed the sensibility of the experiments.

**Review Assessment: Thoroughness In Paper Reading:**

I read the paper at least twice and used my best judgement in assessing the paper.

---

> ### Author Response · Authors · 2019-11-15
> **Reply to review #2**
>
> We thank the reviewer for this constructive and thoughtful feedback.
> “This undermines the *bold* claim in the abstract”: The remark about the bold claim is a fair point, and we have updated the paper with this caveat accordingly.
>
> “This paper raises the issue of ill-definedness of “early in training”, but did not provide a solution.“: We agree that the fact that we do not provide a solution to the problem of late resetting is slightly disappointing. Yet, this is not the main focus of our paper.
>
> “the ability to exactly perserve the accuracy while pruning the weights“: We emphasize that our primary aim is to better understand lottery tickets rather than just get good performance. In particular, we are interested in whether the winning ticket initializations derived from data with little or no supervision outperform subnetworks initialized randomly. Our finding that these winning tickets do in fact outperform random tickets suggests that the properties of winning ticket initializations which lead to better optimization are largely independent of labels, and rather mostly rely on p(x) (though we do note, as the reviewer pointed out, that the inclusion of labels does lead to better winning tickets, though not by much).
>
> “I feel that the novelty of this paper is limited, and do not provide much new insights.“”:  Please see our general comment for more detail on the novelty of our work and why the insights we generated are relevant to future work on the lottery ticket hypothesis.

---

### Author Response · Authors · 2019-11-15
**Global comment**

We thank the reviewers for taking the time to provide detailed and thoughtful comments. This constructive feedback has been helping us improving our submission.

Our contribution is essentially experimental and we were pleased to see that overall, the reviewers found our experimental results to be “solid and provide more understandings of the lottery ticket hypothesis” (R2) and assessed that we have “conducted extensive experiments on three open questions and results prove [ours] assumptions” (R4).
Yet, reviewer 1 is concerned by the robustness of our experimental setup and we address his or her concerns in our reply and in the updated version of the paper.

The main caveat from the reviewers relates to the lack of novelty (R2: “the novelty of this paper is limited”; R3: “immediate followup on Morcos et al.”, “fairly obvious”). They also question the interest and practical value of our study (R2: “do not provide much new insights”; R3: “ I also don't see a practical benefit”).

*Novelty.*
To the best of our knowledge, we propose the first study of the lottery ticket hypothesis in the context of limited access to samples and labels*.* Our experiments are fairly extensive: we generate winning tickets on ImageNet for several different settings (2 different self-supervision losses, 4 different sizes of dataset and 4 different number of classes, semi-supervision) at 14 different pruning rates ranging from 20% to 99.9%, thus covering both moderate and extreme sparsity.
We are the first paper addressing the lottery ticket hypothesis with a majority of our experiments conducted on ImageNet,
while showing that conclusions on smaller datasets may be misleading. Our experiments show indeed that deep networks trained on CIFAR-10 are naturally sparse, making conclusions potentially incorrect.

Moreover, our findings are different from Morcos et al., who show that winning tickets can transfer between different datasets with a common domain (natural images) trained on the same task (labels classification). The fact that we can transfer between similar supervised tasks does not suggest that we should be able to transfer from self-supervised to supervised tasks. Also, it does not guarantee that winning tickets found with only 10 classes (out of 1000) transfer well to full ImageNet for example. Besides, even if these results were expected somehow, it would still be essential to verify these with rigorous experiments, as we propose in our paper.


*Motivation - why does it matter ?*
In our submission, we aim to better understand the properties of winning tickets. Indeed, a better understanding of winning ticket properties might enable faster winning ticket generation and thus allow for concrete applications in fields such as network compression or initialization. We propose an extensive series of experiments investigating winning ticket generation with limited access to labels and samples in order to isolate and assess the dependance in p(x) and p(y|x) of the winning tickets.
As a byproduct of this design, the label-agnostic winning tickets also allow to study the transferability of winning tickets between different tasks, which has a concrete practical benefit. Indeed, similar to the motivation of Morcos et al., if winning tickets can transfer between tasks, then they can be reused across a variety of problems, thus dispensing the need for generating new winning tickets for each new task.

---

### Decision · Program_Chairs · 2019-12-19

**Decision:**

Reject

**Comment:**

The paper studies finding winning tickets with limited supervision. The authors consider a variety of different settings. An interesting contribution is to show that findings on small datasets may be misleading. That said, all three reviewers agree that novelty is limited, and some found inconsistencies and passages that were hard to read: Based on this, it seems the paper doesn't quite meet the ICLR bar in its current form.